# App-Based Rehabilitation in Back Pain, a Systematic Review

**DOI:** 10.3390/jpm12101558

**Published:** 2022-09-22

**Authors:** Claire Stark, John Cunningham, Peter Turner, Michael A. Johnson, Henrik C. Bäcker

**Affiliations:** 1Department of Orthopaedic Surgery, Royal Melbourne Hospital, 300 Grattan Street, Parkville, VIC 3050, Australia; 2Epworth Richmond Hospital, 89 Bridge Road, Richmond, VIC 3021, Australia

**Keywords:** applications, back, spine, pain, rehabilitation

## Abstract

Smartphones and their associated applications are used frequently by patients and clinicians alike. Despite the technology being widely accessible, their use to aid in rehabilitation is yet to be adopted. The SARS CoV-2 pandemic has presented an opportunity to expedite their integration given the difficulty patients currently have in accessing healthcare. The aim of this study was to perform a systematic literature review on the use of smartphone rehabilitation applications compared to standard physiotherapy for back pain. We conducted a search of Medline/Pubmed and google databases using the search terms [APP] AND [[Orthopaedic] OR [Neurosurgery]], following the PRISMA guidelines. All prospective studies investigating rehabilitation applications for back pain or following spine surgery were included. A total of nine studies met the inclusion criteria which investigated 7636 patients, of which 92.4% were allocated to the interventional group (*n* = 7055/7636) with a follow up of 4 weeks to 6 months. All except one study reported on patients experiencing back pain on average for 19.6 ± 11.6 months. The VAS-pain score was presented in all studies without significance between the interventional and control group (*p* = 0.399 before and *p* = 0.277 after intervention). Only one research group found significantly higher improvement in PROMs for the application group, whereas the remaining showed similar results compared to the control group. Using application-based rehabilitation programs provides an easily accessible alternative or substitute to traditional physiotherapy for patients with back pain. Given that smartphones are so prevalent in activities in our daily lives, this will enhance and improve rehabilitation if patients are self-dedicated and compliant.

## 1. Introduction

Rehabilitation is a common feature in patients in the post-operative phase after spinal surgery as well as patients with chronic back pain [1]. Traditionally, these rehabilitation services have been delivered through face-to-face consultation with patients. Since the advent of SARS-CoV2, there has been a prompt turn to the digitalisation of the provision of healthcare [2]. The pandemic has highlighted the advantages of remote rehabilitation programs delivered through a smartphone device.

Smartphone ownership worldwide surpassed 6 billion in July 2022 [3]. In 2021 in the United States, over a third of the population’s media time was spent on mobile phones, and 72.3% of that was on smartphones [4]. The significant utilisation of smartphones and their apps provides an opportunity to integrate their use into clinical practice and help reduce the barriers patients face in accessing health care.

Apps are increasingly used in healthcare, streamlining communication, recording patient outcome data and in some cases measuring outcome data. A survey of 146 patients in a neurosurgical waiting room found that 81% of patients (whom had not had previous surgery) expressed interest in using a postoperative communication and monitoring app [5]. A 2015 study found that there were 72 individual spine surgery-themed apps, of which 45 were free to download; however, only 56% had named medical professionals involved in their development or content [6].

There is evidence supporting telerehabilitation in general orthopaedics [7]; however, there is a void when specifically referring to app-based rehabilitation in back pain and following spine surgery. This systematic review aims to summarize the existing literature and data reporting the outcomes of app-based rehabilitation programs in back pain and following spine surgery.

## 2. Materials and Methods

A systematic review was performed on the 30 July 2022. The Pubmed/MEDLINE, Cochrane and Google Scholar databases were searched following the Preferred Reporting Items for Systematic Reviews and Meta-Analyses (PRISMA) guidelines [8]. The search terms included [APP] AND [[Orthopaedic] OR [Neurosurgery]], which were thought to be the broadest terms. All studies were included that presented their results in English, German or French, analysing the outcome of smartphone app-based rehabilitation in back pain patients and those following spine surgery. Non-accessible full articles, letters to the editors and comments were excluded, as well as those which failed to present functional outcome following rehabilitation.

Applying the PICO scheme, our objective included the comparison of control versus interventional group (O, C) in patients with low back pain (P, I). Hereby, we assumed that the app-based rehabilitation is at least as efficient as general physiotherapy (C, O).

The quality of publications as well as the risk of bias was assessed (Table 1). Data on population demoraphics such as including age, gender, duration of back pain, body mass index (BMI), indication, follow up, patient reported outcome measures (PROM) and apps used were recorded. As functional outcomes, the visual analogue scale of pain (VAS), SF-36, Likert score, PHQ-9, Korff and current symptom score (CSS) were used. In addition, the significances presented in the individual studies were noted, comparing the control with the intervention group.

SPSS (SPSS, Inc., IBM Company, Chicago, IL, USA) and Microsoft Excel (Microsoft Corporation, Redmond, WA, USA) were applied for statistical analyses. Data are presented as absolute numbers and percentages and significances are set to *p*-values < 0.05.

Using our search terms, a total of 1122 articles were found and screened for inclusion criteria. Overall, 91 articles were duplicates and a total of 105 articles were screened. One article needed to be exluded as it analyzed postoperative recovery without specifying the surgery [9]. In total, 9 articles were included in the final analysis (Figure 1).

## 3. Results

Within the 9 prospective studies, 7636 patients were investigated, of which 466 were assigned to the control group. Irvine AB et al. also included an alternative group (*n* = 199), leaving 7055 patients in the interventional group. The mean age was 44.2 ± 7.4 years, and the majority of patients were females (75.3%, *n* = 5638/7487). Where BMI was included, the mean was 26.3 ± 2.2 kg/m^2^, and the pain duration reported was 19.6 ± 11.6 months. All findings are presented in Table 2.

Some authors reported on chronic lower back pain, others reported on back and neck pain [10], and lastly on non-specific back pain [11]. All studies lacked detailed definitions, raising questions regarding the aetiology of the aforementioned pain.

Smartphone applications included the Kaya App [12,13], Snapcare [14], Fitbit app [15] or FitBack [16]. In the remaining studies, the app used was not specified. Follow up varied from 4 weeks [11] to 6 months [15]. Not only did the follow up presentation differ between the groups, but also the presentation of the results. 

The only consistent patient reported outcome measure was the visual analogue scale of pain at rest, which was 4.9 ± 1.2 for the interventional group and 5.2 ± 1.2 for the control group. This improved in the long term to 3.1 ± 1.0 and 3.6 ± 0.5, respectively. However, no significant differences were found between the two groups (*p* = 0.399 before and *p* = 0.277 after intervention). 

Seven authors described significant findings (*p* < 0.05). This varied between the pain, vitality [17], physical function and Oswestry score or overall [12,14,16,18]. Irvine et al. further described significant differences between the control and treatment group after 16 weeks. Bailey et al., Huber S et al. and Amorin AB et al. did not report any significant differences. All findings are illustrated in Table 3 and Table 4.

## 4. Discussion

This systematic review shows no significant differences between application-based rehabilitation and standard physiotherapy (control group) in patients who suffer from back pain for a mean of 19.6 ± 11.6 months. In most studies, the pain improved significantly despite the technique of rehabilitation. Due to the heterogeneity of data, a true meta analysis could not be executed.

Applications in healthcare currently include diabetes [19], weight loss [20], mental health [21], speech disorders [22] and cardiovascular diseases [23], which need to be assessed according to the content quality and benchmark the interventions against best practice guidelines.

Adherence to a postoperative rehabilitation program is one of the major barriers to successful app-based rehabilitation [24]. The compliance is typically low and up to 30% fail to attend classes [25,26]. Consistency in program engagement is crucial to achieving a satisfying outcome. In self-motivated patients with high compliance, app-based rehabilitation shows an effective approach for pain improvement. Rather than presenting different exercises, a sensor could be used to give live feedback to the patients, such as measuring the muscle strength applied.

Within the investigated studies, a variety of different apps were used. The Kaya App adopts comprehensive evidence-based multidisciplinary pain treatment following the international disease management guidelines according to the authors [12,13]. Further significantly lower pain intensity scores were found compared to the control group. The app could also be used during the waiting time until patients are admitted to the pain clinic, as it seems to be an effective low-cost treatment without delay [12]. In contrast, the Snapcare app was designed to monitor the patient’s daily activity level and symptomatic profile. Thereby, individual home exercises are presented and individual activity goals set. These are selected based on the baseline health data, PROM scores and pain levels which are assessed after each activity session [14]. Likewise, Fitbit monitors the individual goals and physical activates and report on physical activity-related goals. In addition, a health coach gives regular feedback via telephone and able to discuss the participants goals and progress. Further individual healthy tips are provided to the users [15]. According to Irvine et al., FitBack is an online app which provides self-monitoring of cognitive and behavioural strategies to improve self-care and back pain-prevention behaviours Exercises are selected based on safety with minimal equipment which can be performed without supervision [16].

Machado et al. performed a search and found a total of 61 available apps in 2016. The majority offered a combination of biomechanical exercises, yoga or strengthening/stretching. Those which scored the highest number of points recommended a combination of biomechanical exercises including strengthening, stretching, core stability or McKenzie exercises [27]. One weakness outlined was the questionable evidence-based intervention, as the majority had not been tested in a randomized controlled trial. Additionally, the authors mentioned that the app quality did not correlate with the in-app or online user ratings. Therefore, they concluded that the user ratings are invalid indicators of app quality. This may relate to a missing pre-exercise questionnaire assessing preconditions such as comorbidities or previous surgeries. Further, the users may have different experience levels, which should be considered.

There are further considerations to implementing app-based rehabilitation programs in community healthcare. The cost of app download was not mentioned, with some applications requiring a single payment for download, whereas others require a subscription-type model. Additionally, whilst smartphones are prevalent in the general community, the usability and app interface would need to consider the target audience. Finally, whilst an app-based rehabilitation is an exciting development in digital healthcare, the safety of those engaging in an unsupervised activity needs to be forefront of mind. An app would need to consider the risk of certain exercises (i.e., falls) if undertaken alone.

The largest cohort investigating the impact of app-based rehabilitation on back pain included 6468 patients. The authors reported a high completion and engagement rate, providing benefits for the groups. The average improvement in VAS pain was 68.5% within the first 12 weeks, where 78.6% completed the program regularly. For back pain, the standardized mean difference was 1.37, which was the same for both genders. Unfortunately, the study failed to include a control group, and since it was a longitudinal observational study, no detailed findings were presented [10].

There are several limitations to this study. In the search terms, we did not include [physical therapy], as we believed that this would return articles relating to unspecific back pain or nutritional apps. The quality of the individual studies was low (range of bias scores 1–3/5), and therefore a meta-analysis was not completed due to the heterogeneity of the data. However, these studies represent the most important examples in this field. The consistent factor within the studies was the visual analogue scale of pain. Additionally, the time to follow up ranged from only 4 weeks to 6 months. Furthermore, different patient reported outcome measures were used, including SF-36, Likert, the Oswestry Disability Index, current symptoms, PHQ-9, and the Korff score. Finally, it has to be mentioned that chronic back pain may resolve itself over time regardless of the rehabilitation activities performed. However, we would expect a significant difference between the two groups, as rehabilitation activities may hasten the rehabilitation.

## 5. Conclusions

Application-based rehabilitation for back pain and following spine surgery is as good as standard physiotherapy. Although no significant differences can be found between the two cohorts, application-based rehabilitation’s integration into healthcare seems promising, especially in motivated patients who regularly engage in independent rehabilitation. Furthermore, in patients who are unable to visit physiotherapists, such as during pandemics or due to living in rural locations, this is an excellent approach which may further lower healthcare costs.

## Figures and Tables

**Figure 1 jpm-12-01558-f001:**
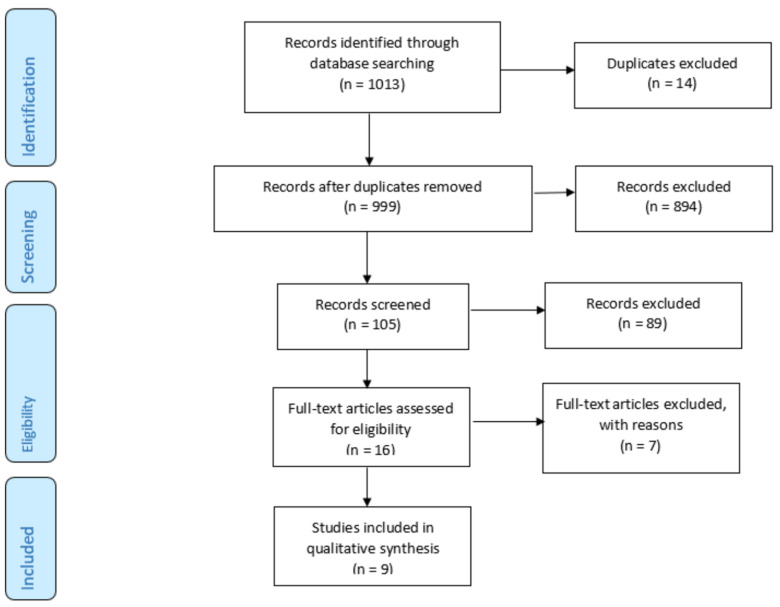
Included articles according to the PRISMA Guidelines.

**Table 1 jpm-12-01558-t001:** Quality assessment results using the risk-of-bias assessment tool.

First Author	Year of Publication	Randomization	Allocation Concealment	Incomplete Outcome Data	Adequate Follow Up	Selective Reporting
Amorim AB	2019	+	-	-	-	-
Bailey JF	2020	-	-	+	-	-
Chhabra HS	2018	+	-	+	-	+
Hasenöhrl T	2020	-	+	+	-	+
Huber S	2017	-	-	-	-	-
Irvine AB	2015	+	-	-	-	+
Shebib R	2019	+	+	+	-	-
Toelle TR	2019	+	+	-	-	-
Yang J	2019	+	-	+	-	+

**Table 2 jpm-12-01558-t002:** Demographics of all included study.

First Author	Year of Publication	Intervention	Indication	Number of Patients (*n*)	Age (Years)	Gender (Female)	Bodyweight (kg)	BMI (kg/m^2^)	Pain Duration (Months)
Amorim AB	2019	Fitbit app	Chronic low back pain	31	59.5 ± 11.9	15		28.9 ± 6.0	
		Control		24	57.1 ± 14.9	19		27.2 ± 5.1	
Bailey JF	2020	Unspecified app	Neck and Backpain	6468	42.6 ± 10.9	4981		29.8 ± 7.1	
Chhabra HS	2018	Snapcare app	Chronic low back pain	45	41.4 ± 14.2		63.4 ± 12.5	23.2 ± 4.2	22.8 ± 22.0
		Control		48	41.0 ± 14.2		66.2 ± 11.5	23.5 ± 3.8	28.0 ± 25.5
Hasenöhrl T	2020	Unspecified app	Non specific back pain	27			81.7 ± 22.5	28.1 ± 7.1	
Huber S	2017	Kaya app	Low back pain	105	33.9 ± 10.9	105			More than 12 weeks (73.3%)
Irvine AB	2015	Fitback	Low back pain	199		116			
		Alternative care		199		117			
		Control		199		125			
Shebib R	2019	Unspecified app	Back pain	133	43.0 ± 11.0	37%		26.0 ± 5.0	
		Control		64	43.0 ± 12.0	48%		26.0 ± 4.0	
Toelle TR	2019	Kaya App	Chronic low back pain	42	41.0 ± 10.6	35		24.4 ± 3.3	7.2 ± 3.4
		Control		44	43.0 ± 11.0	31		25.4 ± 4.6	6.7 ± 3.1
Yang J	2019	unspecified app	Chronic low back pain	5	35.0 ± 19.3	1	64.8 ± 10.3		35.8 ± 54.4
		Control		3	50.3 ± 9.3	3	62.0 ± 15.9		17.0 ± 17.1
Sum				7636		5548			
Average					44.2 ± 7.4		67.7 ± 7.2	26.3 ± 2.2	19.6 ± 11.6

**Table 3 jpm-12-01558-t003:** Indication, Oswestry score and VAS of pain; overall means, standard deviations and *p*-values. R-VAS—visual analogue scale of pain at rest, A-VAS—visual analogue scale of pain during activity, LT—long term, ST—short term.

First Author	Year of Publication	Intervention	Follow Up	ODI Score Before	ODI Score ST	ODI Score LT	R-VAS	ST	LT	A-VAS	ST	LT	Significances
Amorim AB	2019	Fitbit app	6 months				5.3		3.8				*p* = 0.815
		control					5.1		4.0				
Bailey JF	2020	Intervention	12 weeks				4.6		1.4				
Chhabra HS	2018	Snapcare	12 weeks				7.3		3.3				***p* < 0.001**
		Control					6.6		3.2				***p* < 0.05**
Hasenöhrl T	2020	Unspecified app	4 weeks	17.1	14.4		3.2	3.2					
Huber S	2017	Kaya App	12 weeks				4.8	3.2	2.6				*p* < 0.001
Irvine AB	2015	Fitback	16 weeks				3.0	3.3	3.4				***p* < 0.001, between control and treatment**
		Alternative care					3.0	3.3	3.5				
		Control					2.9	3.1	3.3				
Shebib R	2019	Unspecified app	12 weeks	21.7		19.7	4.6		4.4	3.9		3.7	
		Control		21.0		18.9	4.5		4.3	4.4		4.1	***p* < 0.05**
Toelle TR	2019	Kaya App	12 weeks				5.1	4.3	2.7				
		Control					5.4	4.1	3.4				***p* = 0.021**
Yang J	2019	Unspecified app	4 weeks				5.9	3.4					***p* < 0.05 for vitality**
		Control					6.0	6.0					
Sum		Intervention					4.9 ± 1.2	3.5 ± 0.5	3.1 ± 1.0	3.9		3.7	
		Control					5.2 ± 1.2	4.4 ± 1.5	3.6 ± 0.5	4.4		4.1	

**Table 4 jpm-12-01558-t004:** PROMs following rehabilitation programs for back pain and following spine surgery. Score indicates the PROM used; overall means, standard deviations and *p*-values.

First Author	Year of Publication	Score	Pain	ST	LT	Symptoms/Emotions/Other	ST	LT	Function in ADL	ST	LT	Sport/Recreation	ST	LT	Quality of Life/Vitality	ST	LT	Overall	ST
Amorim AB	2019	Likert										202.20		187.70	1984.90		2065.70		
												200.50		169.20	1936.70		1941.20		
Bailey JF	2020	PHQ-9/Korff	15.95		7.75	4.39						3.35			11.56				
Chhabra HS	2018	Current Symptom score/SF-36	7.02	3.27		2.11	1.22		4.82	3.02		2.58	1.27		2.09	1.04		52.1	20.2
																		41.4	29.2
Hasenöhrl T	2020	SF-36	38.78	53.59		71.26	80.25		65.15	68.41		72.78	77.78		54.44	61.67			
Huber S	2017	VAS																	
Irvine AB	2015	Multidimensional Pain Inventory Interference Scale. Dartmouth CO-OP. WLQ	2.96	3.32	3.38	4.02	4.59	4.90	3.83	3.27	3.03				3.14	3.38	3.51		
			3.01	3.30	3.47	4.07	4.48	4.65	3.93	3.45	3.31				3.10	3.34	3.37		
			2.92	3.08	3.28	4.08	4.03	4.12	4.03	3.85	3.74				3.09	3.11	3.14		
Shebib R	2019	VAS																	
Toelle TR	2019	SF-36	45.53	41.65		44.38	46.53								48.69	50.58			
			47.32	40.78		44.56	45.56								47.64	48.64			
Yang J	2019	SF-36	44.00	40.00		58.40	60.07		49.00	50.00		74.00	59.00		50.00	47.00			
			63.33	56.67		66.67	44.56		58.33	65.00		46.67	51.67		63.33	65.00			
Sum		Intervention	42.77 ± 3.54	45.08 ± 7.42		58.01 ± 13.44	62.28 ± 16.97		57.08 ± 11.42	59.21 ± 13.02		73.39 ± 0.86	68.39 ± 13.28		51.04 ± 3.01	53.08 ± 7.65			
		Control	55.33 ± 11.32	48.73 ± 11.25		55.62 ± 15.63	45.06 ± 0.71		58.33	65.00		46.67	51.67		55.49 ± 11.09	56.82 ± 11.57			

## Data Availability

Not applicable.

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
