# Peer review of "App-Based Rehabilitation in Back Pain, a Systematic Review"

_jpm, 2022, doi:10.3390/jpm12101558_

Round 1

Reviewer 1 Report

Findings:

If all but 1 study showed non-significant differences between control and app groups, doesn't that mean that app rehabilitation does not work? Why are the conclusions: "Application based rehab for back pain and following spine surgery is as good as standard physiotherapy."? This does not make sense. Was the control standard physiotherapy in the located studies?

Table 1:

What was the bias /quality assessment tool used? i.e. SIGN-50, NOS, JBI etc.?

Further, no discussion states that the quality of the studies was low (range of bias scores was 1-3/5). Therefore, regardless of findings, no conclusion can be made as the quality is overall low in the 10 located studies.

Figure 1:

The PRISMA flow chart numbers do not add up!!! The last paragraph in the Methods section and boxes in flow chart do not make sense.

Table 2:

Please add standard deviations.

Also, for last row state sum OR avg.

Discussion:

Even though 1 paper (Bailey) contributed an n of 6468, no specific discussion is given to the weight of that study. 

First paragraph - "in most studies, the pain improved significantly despite the technique of rehabilitation" - Is this not because acute back pain likely resolves over time, or is it due to the waxing and waning of chronic low back pain? Again, if the controls (in included studies) were untreated and no differences were observed, both groups improved even without treatment? This would support the fluctuating nature of chronic pain.

Minor edits:

Line 34: "through and mobile..." - awkward

Line 63,64: - awkward

Line 68: Change "population such as demographics" to "population demographics such as..."

Line 156: "did not necessarily correlated with ..." - awkward

Line 171: "back pain may varied..." - awkward

Line 182: Change "which further may..." to "which may further..."

Author Response

Thank you so much for your time and effort in your comments. Please find our detailed response in the attached document.

Reviewer 2 Report

Thank you for inviting me to review manuscript "App based rehabilitation in back pain, a systematic review". The authors reported a systemic review consisted of 10 studies, comparing App-base rehab (7139 patients) to traditional physiotherapy (665 patients) in the treatment of chronic back or neck pain (9 studies) or postoperative back pain (1 study). This is an interesting topic but my comments and concerns are as follows.

Page 1 Line 18: Please be consistent on the digits after decimal point,

eg., 19.6 ± 11.6 months, instead of 19.59±11.6months.

Please check the whole manuscript for correction. 

Page 2 Line 56: If the majority of the studies included were treating chronic back pain instead of postoperative pain after spinal surgery, why not include [Physical therapy] in your search terms?

Page 4 Table 2: the “Indication” and “Group” columns were obviously misplaced.

Page 5 Line 95: which of the study report on treating “back and neck pain”? Please define in Table 2 

Page 6-8 Table 3:

-              Instead of “Indication”, isn’t “Intervention” more suitable as the heading of the third column? Please also revise the contents of this column into “Fitbitt app”, “control”, “Snapcare app”, and “unspecified app” etc., as proper. The app “Fitback” used by Irvine et al. was not mentioned in Line 98. Was it different from “Fitbitt app” used by Amorim? Please define.

-              Also, please define your abbreviations, what is “R-VAS”, “A-VAS”“LT”, and “ST”?

-              The presentation of p values was quite confusing. Do the values p > 0.30 or p > 0.05 have a specific number? The p values of a few studies were also < 0.05 (Shebib et al, Toelle et al, and Yang). Were they not pointing out a significant difference from the control group? 

Line 134: “A mean of” 19.59 ± 11.6 months, instead of “approximately”. 

Line 141: The study of Hou et al was the only study which included postoperative patient. This caused heterogeneity in terms of data interpretation. In addition, the study of Hou et al included patients with a wide spectrum of diagnosis, consisted of lumbar disc herniation, spinal stenosis, or lumbar spondylolisthesis. The inclusion of the study was rather heterogeneous by itself. Wouldn’t it be proper to exclude this study from your analysis?

Introduction and Discussion: The authors should perhaps introduce to us what was the contents of these Apps. And if one of them led to a better outcome, what was the difference of it comparing to all other applications.

Author Response

Many thanks for your thorough review. We adjusted our manuscript accordingly. 

Round 2

Reviewer 1 Report

The significant changes made improved the paper greatly.

Minor edits:

Line 94: "within the 10 prospective studies..."  - (change to 9)

Line 186: "correlated" - (change to 'correlate')

Line 214: "24 months" - (change to '6 months')

Author Response

Thank you so much. We made all effort to comply with the comments and changed the outlined points accordingly.

Reviewer 2 Report

Thank you for all the hard work during your revision. 

I have no further questions.

Author Response

Thank you so much for the comments. 

No changes were required.